# Multi-Agent Optimal Control for Central Chiller Plants Using Reinforcement Learning and Game Theory

**Shunian Qiu [1], Zhenhai Li [2], Zhihong Pang [3], Zhengwei Li [2] and Yinying Tao [4,\*]**

1   School of Civil Engineering and Architecture, Zhejiang University of Science and Technology, Hangzhou 310023, China
2   School of Mechanical Engineering, Tongji University, Shanghai 200092, China
3   Department of Construction Management, Louisiana State University, Patrick F. Taylor Hall 3315-D, Baton Rouge, LA 70803, USA
4   School of Design and Fashion, Zhejiang University of Science and Technology, Hangzhou 310023, China
\*   Correspondence: tao@zust.edu.cn

**Abstract:** To conserve building energy, optimal operation of a building's energy systems, especially heating, ventilation and air-conditioning (HVAC) systems, is important. This study focuses on the optimization of the central chiller plant, which accounts for a large portion of the HVAC system's energy consumption. Classic optimal control methods for central chiller plants are mostly based on system performance models which takes much effort and cost to establish. In addition, inevitable model error could cause control risk to the applied system. To mitigate the model dependency of HVAC optimal control, reinforcement learning (RL) algorithms have been drawing attention in the HVAC control domain due to its model-free feature. Currently, the RL-based optimization of central chiller plants faces several challenges: (1) existing model-free control methods based on RL typically adopt single-agent scheme, which brings high training cost and long training period when optimizing multiple controllable variables for large-scaled systems; (2) multi-agent scheme could overcome the former problem, but it also requires a proper coordination mechanism to harmonize the potential conflicts among all involved RL agents; (3) previous agent coordination frameworks (identified by distributed control or decentralized control) are mainly designed for model-based control methods instead of model-free controllers. To tackle the problems above, this article proposes a multi-agent, model-free optimal control approach for central chiller plants. This approach utilizes game theory and the RL algorithm SARSA for agent coordination and learning, respectively. A data-driven system model is set up using measured field data of a real HVAC system for simulation. The simulation case study results suggest that the energy saving performance (both short- and long-term) of the proposed approach (over 10% in a cooling season compared to the rule-based baseline controller) is close to the classic multi-agent reinforcement learning (MARL) algorithm WoLF-PHC; moreover, the proposed approach's nature of few pending parameters makes it more feasible and robust for engineering practices than the WoLF-PHC algorithm.

**Keywords:** central chiller plant; game theory; model-free control; multi-agent reinforcement learning; agent-based control; multi-agent system





## 1. Introduction

With the development of modern society, people are spending more time in buildings. In 2022, the buildings and building construction sector accounted for 30% of global energy consumption and 27% of total energy emissions [1]. Considering the urgency of environment protection, it is necessary to reduce the energy consumption and carbon emission of the building sector. To realize that goal, appropriate operation of a building's heating, ventilation and air-conditioning (HVAC) is necessary since the HVAC system is one of the major energy consumers in buildings [2].

### 1.1. Optimal Control to Central Chiller Plants

The central chiller plant, as a critical sub-system in HVAC systems and accounts for a substantial part of building energy consumption [3]. A typical central chiller plant is comprised of chiller(s), condenser water pump(s), chilled water pump(s) and cooling tower(s), and the chiller is the key component of this system [4].

In a central chiller plant, usually the controllable variables include chilled water temperature, chiller running number, cooling tower running number, cooling tower fans' working frequency, chilled/condenser water pump running number and frequency [4,5]. In order to conserve building energy, the optimal control/operation of central chiller plants (especially the chillers) has been widely studied for decades [6–10].

Wang et al. [11] established the near-optimal performance map of case chillers by mining the historical operational field data and other supplementary data of the case system. The map is a refined database to describe the relationship between chillers' near-optimal coefficient of performance (COP, representing chillers' energy efficiency) and the working condition, which is comprised of two input arguments: chiller cooling load and the temperature difference between evaporator inlet and condenser inlet. Moreover, the performance map records the historical control plan corresponding to each near-optimal point. Based on the established performance map, the near-optimal system energy efficiency along with its control plan under a given working condition (i.e., system cooling load and temperature difference) could be retrieved. Therefore, in the real-time control phase, their proposed control logic could properly control the system under the current condition to maximize the estimated overall energy efficiency.

Wang et al. [10] proposed an event-driven optimal control strategy using supervised data-mining technology. In the offline preparation phase, their proposed strategy uses a random forest regressor along with the system's operational data, to identify the critical variables which influence the building energy efficiency most. Then, in real-time control phase, when the critical variables change significantly (i.e., an "event" occurs), the optimization controller would execute the pre-defined optimization protocol (based on a system model) to adjust the setpoints of chilled water temperature, condenser water supply temperature and supply air temperature. The simulation case study suggests that, for their case system, part load ratio (PLR) and chilled water flowrate are closely related to system's overall energy efficiency. Moreover, their proposed event-driven optimal control strategy outperforms the time-driven traditional control method by 0.9–4.6% of energy conservation.

The abovementioned two studies both proposed an optimal control strategy on the basis of sufficient historical system operational data because they do need historical data to build a model/map to estimate the systems' energy efficiency corresponding to each potential control plan. In doing so, the optimal controller could adjust the controllable variables with confidence. However, model-based optimal control strategies face some common shortcomings such as model inaccuracy risk and a considerable cost of high-quality data [12–14]. Hou et al. [12] reviewed previous studies on building optimal control and pointed out that, although lots of effort has been devoted to improve model accuracy, model errors exist universally and are impossible to eliminate. When model error exists, the calculated output (optimization target) of the system model differs from the real value, which could misguide the decision-making process of the optimization controller [14].

To tackle the universal problem of model-based control, some researchers adopted reinforcement learning (RL) techniques to carry out model-free optimal control [13,15–17]; the details are address in Section 1.2.

### 1.2. Application of Reinforcement Learning (RL) Techniques in HVAC System Control

Since two decades ago, researchers have started to investigate the application value of RL algorithms in the optimal control of building energy systems (especially HVAC systems) [16–18]. Typically, utilizing RL techniques could be divided into three steps: Step 1, formulate one or more RL agents (defining their action space, state space, reward



function); Step 2, let the agent(s) interact with the virtual/real environment and update agent(s)' experience and policies with selected RL algorithms; Step 3, commission the trained agent in the targeted system for optimal control. It should be noted that, in Step 3, the agent(s) still learn to improve their decision-making policies when working. Hence, Step 2 is not mandatory [13].

Esrafilian-Najafabadi and Haghighat [19] adopted double DQN (deep Q network) to optimize the operation of a HVAC system. They defined the RL agent's state with outdoor and indoor air temperature, occupancy status and time. The setpoint of indoor air temperature is chosen as the RL agent's action; the reward function is composed of heating energy and the temperature difference between indoor air and its setpoint. After being formulated, the RL agent trained itself via double DQN algorithm within a virtual black-box building model derived from EnergyPlus [20]. After training, the performance of the RL agent was tested on a new black-box system model (set up with another test dataset), and results indicate that the RL control method could outperform the model-predictive controller (MPC) in maintaining the indoor thermal comfort.

Apart from simply saving the energy consumption of building HVAC systems, many researchers have been investigating how to save energy costs by utilizing the time-of-use electricity tariff, the thinking of which is also known as demand response or load shifting [21]. To realize load shifting for building cooling systems, Schreiber et al. [22] proposed an RL-based control method to adjust the set points of chilled water temperature, chilled water flowrate and valve positions. They compared the performance of DQN and DDPG (deep deterministic policy gradient) on this task. The simulation case study via Modelica shows that DDPG could save about 14% energy cost compared to the baseline (purely demand-oriented strategy). Moreover, DDPG performs slightly better than DQN on this load shifting task.

Liu et al. [23] presented a multi-step RL control method based on DDPG algorithm and predictive models. In that study, the time sequence of outdoor air temperatures and the real-time electricity price compose the state variables; the output power of the case HVAC system is taken as the agent's action; and reward function is specified considering energy consumption and user comfort. To acquire future outdoor air temperature for RL agent's state recognition, they modified the long short-term memory (LSTM) network with generalized co-entropy loss to better forecast future outdoor temperature. Their proposed method conserved over 10% more HVAC energy than on–off control conserved in simulations.

RL-based control is also promising in its use to optimize large-scaled district cooling systems. Wang et al. [24] adopted Q-learning to optimize the set points of chilled water pressure of two cold sources. In that study, the RL controller observes valve positions, hydraulic imbalance rate, real-time water pressure set point and system load ratio as state variables, based on which, the controller would adjust the water pressure set point in a stepwise way. Simulation results indicate that their succinct method could shorten the control time-lag by five hours compared to the rule-based controller. Moreover, their RL-based controller led to higher valve opening degrees, saving over 12% of pump energy consumption compared to the rule-based controller.

Except for the simulation studies above, some researchers have implemented RL-based controllers to control actual HVAC systems in the real world. Qiu et al. [13] proposed a hybrid model-free chiller control method based on a RL algorithm and expert knowledge. The outdoor air temperature and the system cooling load are regarded as system state variables. The chilled water temperature is controlled by the agent's action to achieve high chiller utility, which is the reward of the RL agent. Their proposed method was successfully implemented on a real HVAC system in Shanghai; application results suggest that the method could reach the same energy saving performance as manual expert control does. In addition, this team has applied the RL-based control method on another real system for the optimization of chilled water pumps [25].

All studies reviewed in this section optimize HVAC system operation with a single RL agent because they only intend to control one signal (no matter the indoor temperature setpoint or HVAC system's output power). Nevertheless, as introduced in Section 1.1, the operation of central chiller plants are influenced by multiple controllable variables, in which scenario, single agent scheme may face challenges of high training cost, long training period and poor initial performance due to the large jointed action spaces [5,26]. To coordinately optimize the whole central chiller plant, a multi-agent scheme is useful, details of which are addressed in Section 1.3.

*1.3. Coordinated Optimization Problem in HVAC Systems*

As mentioned in the end of Section 1.2, when facing multiple controllable variables, a single RL agent may be not suitable for the model-free optimization problem. If single agent scheme is adopted, then its action space would be the jointed space of all controllable variables' solution space, and the size of the jointed space would increase with the geometric series when more controllable variables become involved. This fact would lead to high training cost and long training period of the RL agent for engineering practices [5,26]. Qiu et al. [5] quantitatively compared the energy-saving performance of the single-agent RL scheme and the multi-agent RL scheme on the HVAC optimal control problem. Simulation results show that for a discretized RL algorithm (policy hill climbing), multi-agent scheme could outperform the single-agent scheme in both short-term (initial stage) and long-term (post convergence) performance. Moreover, multi-agent scheme takes a shorter training period to reach convergence than single-agent scheme does. Fu et al. [26] proposed a multi-agent RL control method based on DQN for central chiller plant control. Simulation results validate the faster converging speed of a multi-agent RL scheme. Hence, multi-agent RL scheme is useful and meaningful for the coordinated optimization of central chiller plants [5,26].

Although multi-agent scheme is suitable for the optimization of central chiller plants, the coordination mechanism of agents (also known as distributed control or decentralized control framework) is necessary but not easy to design [27,28]. Specifically, a central chiller plant is usually composed of cooling towers, chillers, condenser water pumps and chilled water pumps. The operational objectives of these appliances are generally consistent, but not completely identical. Typically, the cooling tower and condenser water pump are controlled for high energy efficiency of the whole system [5], chillers are designated to operate at a high efficiency condition while providing sufficient cooling to the user side [13] and the control objective of chilled water pumps are usually more inclined to the user comfort because they are closer to the user side compared to other appliances [25]. Therefore, a potential problem of the multi-agent scheme is how to harmonize the conflict of objectives among all control agents, coordinating their own sub-control procedures.

To tackle the coordination problem in multi-agent HVAC control field, some researchers have designed different decentralized model-based optimization schemes. Li et al. [29,30] proposed an event-driven, multi-agent optimization framework. Their proposed framework categorizes the whole control system into one PAU (primary air-handling unit) agent, one central coordinator and several room agents. They formulated the comprehensive optimization objective in a centralized way considering the indoor pollutant level and the HVAC system's energy consumption simultaneously. Then they decomposed the original optimization problem into sub-problems via ADMM (alternating direction method of multipliers); all sub-problems would be assigned to underlying agents. All agents are designed to not only optimize the operation of its own region but also consider the constraints. In real-time control, the PAU agent and the room agents solve their own local optimization problems, reporting results to the central coordinator, and the coordinator would update parameters for the next trial of local optimization. In every control time step, iterations would be conducted between the coordinator and underlying agents until a convergence is reached.

Apart from the multi-agent scheme above (regarding individual rooms as agents), some researchers take HVAC devices (especially central chiller plant devices) as agents. Li et al. [31] designed a double-layer multi-agent system (MAS) to optimize the operation of central chiller plants. In their scheme, the whole central chiller plant is divided into four sub-systems: chillers, chilled water pumps, cooling towers and condenser water pumps. The upper layer of the MAS is designed to coordinate the operation of these four sub-systems (e.g., all sub-system agents would "negotiate" to determine a proper total chilled water flowrate). Meanwhile, the underlying layer, which is composed of several device agents, focuses on how to assign the operation task above to HVAC devices within each sub-system (e.g., flowrate allocation among chilled water pumps). To realize the coordination among sub-systems and allocation among devices, they proposed a communication topology for agents to interact; game theory was adopted to deal with the conflicts of device agents during the local optimization process; mathematical performance models of chillers and pumps are required for local optimization.

Moreover, some studies expanded the scale of the MAS to the whole HVAC system along with conditioned rooms. To optimize the operation of the whole HVAC system, Wang et al. [28] proposed a hierarchical MAS composed of zone agents (controlling indoor environment setpoints), component agents (controlling chillers, pumps, cooling towers, air-handling units) and one central coordinator agent. In their proposed method, each zone agent is intended to maximize the indoor comfort of its corresponding zone, and a component agent would try to minimize the energy consumption of its corresponding HVAC appliance. The iteration between component/zone agents and the coordinator agent would result in an optimal solution for the whole MAS.

Studies reviewed above [28–31] tried to solve the multi-agent optimization problems in HVAC systems with novel communication topology and coordination mechanisms. However, they still depend on pre-defined device models to function, and only a few studies investigated how to optimize HVAC operation using a multi-agent scheme without models, which is also known as multi-agent reinforcement learning (MARL) [26,32].

Fu et al. [26] established a MAS composed of one chiller agent, several cooling tower agents and condenser water pump agents to optimize the operation of the central chiller plants with a MARL scheme. In their study, the chiller agent is intended to optimize chiller loading of the case system while cooling tower agents and condenser water pump agents are responsible for the optimization of device frequencies. They adopted DQN as the basis learning algorithm for all agents. Agents share a common reward function but do not communicate with each other, which means their agent coordination was realized in a simple, implicit, autonomous way. According to Ref. [33], this "common reward" mechanism could lead to global optimum. Simulation results indicate that the control method proposed by Fu et al. reaches convergence much faster than the single RL agent controller.

Let us summarize this section: (1) the multi-agent scheme has some advantages over the single-agent scheme for model-free RL control of HVAC systems; (2) there are existing studies investigating how to optimize HVAC systems with decentralized MAS, but they still rely on device/system models to function [28–31]; (3) only a few preliminary studies based on MARL tried to solve the multi-agent optimization problem of HVAC systems without models [5,26], but they have not considered the issue of potential objective inconsistency among agents.

### 1.4. Motivation of This Research

As abovementioned, the optimal operation of central chiller plants is important for building energy conservation. To realize optimal control and avoid model inaccuracy risk, RL techniques have been widely utilized to realize model-free optimal control for building HVAC systems. Regarding the model-free optimization problem for the whole central chiller plant, there are problems remaining: (1) the single RL agent scheme could cause a high training cost and a long training period when optimizing multiple variables in the

whole central chiller plant due to the large jointed action space; (2) multi-agent RL scheme is more suitable to optimize multiple controlled variables simultaneously, but there are potential objective conflicts among multiple appliances/agents that need reconciliation; (3) current multi-agent coordination mechanisms are mostly designed for model-based optimization frameworks. To tackle these problems, this study establishes the optimal control framework of central chiller plants with a MAS; after that, this article proposes an optimal control approach based on multi-agent reinforcement learning (MARL, for model-free learning in MAS) and game theory given its abundant methods in solving conflicts [34,35].

In this paper, Section 2 demonstrates the methodology of the proposed control approach; Section 3 introduces the establishment of the simulation environment for performance verification; Section 4 analyzes and discusses the simulation case study results; and Section 5 concludes the paper.

## 2. Methodology

### 2.1. Overview

*Applicable system:* Small-scaled central chiller plants composed of no more than three identical chillers, identical parallel condenser water pumps and identical cooling towers were used [36].

*Algorithm basis:* SARSA (a classic tabular RL algorithm) and Dominant strategy underlining method (a simple method to solve the Nash equilibrium in bilateral matrix games) were used. To apply the abovementioned algorithms, two RL agents are established in Section 2.2: cooling agent (controlling condenser water pumps and cooling towers) and chiller agent (controlling chillers).

*Optimization objectives:* For cooling agent, the objective is the comprehensive system COP concerning chillers, condenser water pumps and cooling towers; for chiller agent, the objective is composed of system *COP* and returned chilled water temperature ($T_{chwr}$) because $T_{chwr}$ could indicate if chillers-supplied cooling is enough to meet the user demand. Details about objective function are addressed in Section 2.2.

*Online and offline preconditions:* A priori knowledge includes historical weather data (ambient wet-bulb temperature), the layout of the case system, nominal characteristics of all chillers, pumps and cooling towers. As for online system monitoring, real-time values of the following variables are required: system cooling load $CL_s$ (kW), returned chilled water temperature $T_{chwr}$ (°C), ambient wet-bulb temperature $T_{wet}$ (°C), total electrical power of the case system $P_s$ (kW).

*Control signals/actions:* Condenser water pump frequency $f_{pump}$ (Hz), cooling tower fan frequency $f_{tower}$ (Hz), setpoint of supplied chilled water temperature $T_{chws}$ (°C) were used.

*Optimization interval:* The proposed method should be executed every 15–30 min because (1) frequent optimal control action could cause oscillation of appliances; (2) larger interval leads to less timely optimization and less energy conservation [37]; (3) the proposed method is based on RL algorithms, which takes environmental reward to update control policy. Hence, the controller must wait for the system to stabilize after the former control interference before its next learning. For small-scaled systems, for which this method is designed, the stabilization could cost approximately 15 min; hence the appropriate optimization interval should be 15–30 min [13].

Figure 1 shows the online workflow of the proposed method which is composed of several steps: rule-based on–off control of all appliances (Section 2.1), optimization of $f_{pump}$, $f_{tower}$, $T_{chws}$ with Nash equilibrium solving (Section 2.3), agent value function updating with SARSA (Section 2.4).

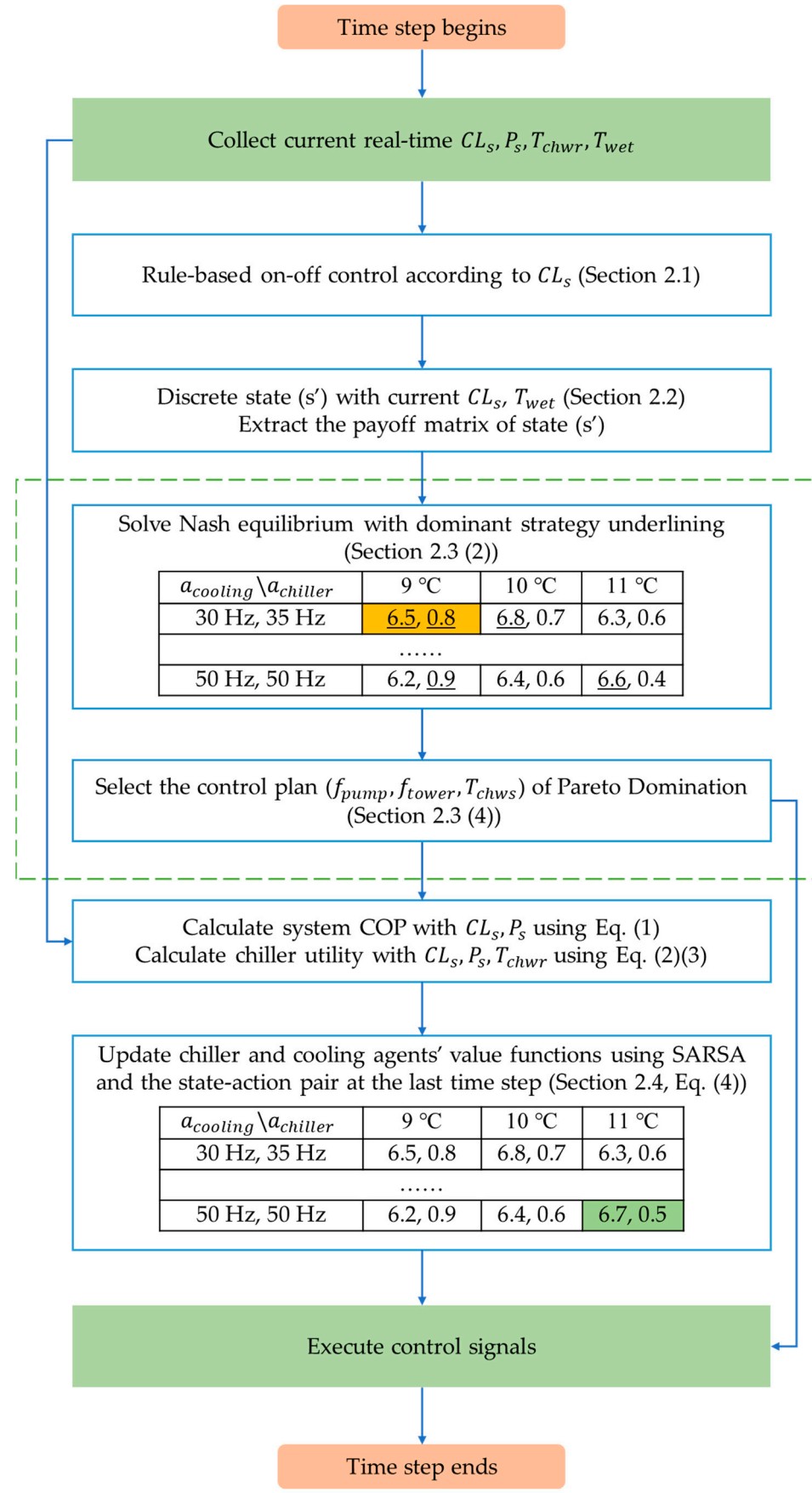

**Figure 1.** Workflow of the proposed control method.

In this study, on–off statuses of all appliances are determined with several simple rules:

(1) The whole central chiller plant only operates when $CL_s$ is over 20% of single chiller's cooling capacity [38–40].

(2) All cooling towers operate simultaneously when the system is on to maximize heat exchange area [41].

(3) Increase chiller running number only when $CL_s$ is larger than current cooling capacity. Shutdown chiller(s) if fewer chillers still meet user's cooling demand [8].

(4) The number of running condenser and chilled water pumps is in accordance with the number of working chillers. Chilled water pump frequency is not optimized in this study.

### 2.2. RL Agent Formulation

In the proposed approach, two RL agents are defined: chiller agent which optimizes $T_{chws}$ and cooling agent which optimizes $f_{pump}, f_{tower}$. Critical variables of these two RL agents are defined as following:

**State:** Two agents share the common state variable, which is composed of discretized system cooling load $CL_s$ (kW) and rounded ambient wet-bulb temperature $T_{wet}$ (integer °C). Note, measured real-time $CL_s$ need to be discretized according to single chiller's cooling capacity. The lower limit of the state space is 20% of single chiller's cooling capacity, and the upper limit is the rated cooling capacity of the whole case system. For instance, if single chiller's cooling capacity of the case system is 1000 kW and the case system consists of three identical chillers, then $CL_s$ needs to be discretized to a space of (200, 300, 400, ... , 3000 kW). Moreover, the upper and lower limits of $T_{wet}$ in state space need to be specified according to the historical weather data of the case system. For example, if $T_{wet}$ of the case system varies within 20–30 °C over the last cooling season, then the upper and lower limits of $T_{wet}$ in state space could be specified as 20/30 °C. An example of the state space is listed in the first line of Table 1.

**Table 1.** Example of the value function based on the payoff matrix.

| 24 °C, 1060 kW | | 25 °C, 1060 kW | | | 28 °C, 2120 kW |
|---|---|---|---|---|---|
| | $a_{cooling} \backslash a_{chiller}$ | 9 °C | 10 °C | 11 °C | |
| *Payoff Matrix$_1$* | 30 Hz, 35 Hz | 6.5, 0.8 | 6.8, 0.7 | 6.3, 0.6 | *Payoff Matrix$_n$* |
| | ... ... | | ... ... | | |
| | 50 Hz, 50 Hz | 6.2, 0.9 | 6.4, 0.6 | 6.6, 0.4 | |

**Action:** For cooling agent, the combination of $f_{tower}$ and $f_{pump}$ are the action variable, whose space is {(30 Hz, 35 Hz), (30 Hz, 40 Hz), (30 Hz, 45 Hz), (30 Hz, 50 Hz), (35 Hz, 35 Hz), ... , (50 Hz, 50 Hz)}. In other words, the alternatives of $f_{tower}$ are (30, 35, 40, 45, 50 Hz), and the alternatives of $f_{pump}$ are (35, 40, 45, 50 Hz) for the safety of case systems [25]. For chiller agent, its action space should be defined according to its nominal $T_{chws}$ value with 1 °C tuning range. For example, if the nominal $T_{chws}$ of the case chiller is 7 °C, then the action space would be (6, 7, 8 °C).

**Reward:** The cooling agent takes the real-time system COP ($COP_s$) as the reward, which is calculated with Equation (1):

$$COP_s = CL_s \div P_s \tag{1}$$

where $P_s$ (kW) is the sum of real-time electrical power of all chillers, condenser water pumps and cooling towers.

For chiller agent, besides the system COP, user side comfort is also worth concern because chiller agent's action $T_{chws}$ directly influences user side comfort and chiller efficiency simultaneously [13]. For an optimization problem with multiple objectives, typically all sub-objectives would be combined with weights to one comprehensive optimization

goal [42]. Hence, utility functions (Equations (2) and (3)) are set up herein to utilize the comfort objective and efficiency objective [13]:

$$U_{comfort} = \frac{1}{1 + \beta_1 exp\left[\beta_2\left(T_{chwr} - T_{chwr,\,ref}\right)\right]} \tag{2}$$

$$U_{chiller} = X_{cop}(COP_s \div COP_{s,nominal}) + (1 - X_{cop})U_{comfort} \tag{3}$$

where $U_{comfort}$ is the comfort utility depending on returned chilled water temperature $T_{chwr}$ (°C) and its reference value $T_{chwr,\,ref}$ (°C) (available on chiller manual), $\beta_1 = 0.25$, $\beta_2 = 4.15$; they are coefficients determining the shape of comfort utility function curve (Figure 2). $U_{chiller}$ is chiller utility as the reward of chiller agent; $COP_{s,nominal}$ is the nominal value of $COP_s$ ($COP_{s,nominal}$ is the quotient of rated system cooling capacity (kW) and rated system electrical power (kW)); $X_{cop}$ is the weight of efficiency objective in chiller utility, and it should be specified within 0–1 according to the need of the applied building. A large $X_{cop}$ would make the chiller controller prefer to increase the $T_{chws}$ set point for better chiller efficiency, rendering a more aggressive control strategy; on the contrary, a small $X_{cop}$ generally leads to a conservative control strategy keeping $T_{chws}$ at a low level to guarantee user comfort. Quantitative analysis about this parameter can be found in Ref. [43]. In this study, it is recommended to set $X_{cop}$ to 0.5 referring to Ref. [13].

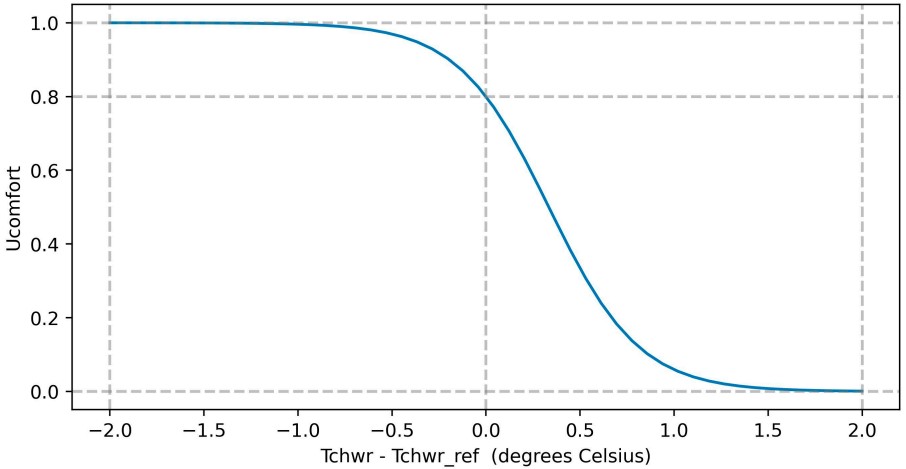

**Figure 2.** Comfort utility function curve ($\beta_1 = 0.25$, $\beta_2 = 4.15$).

*Value function:* Since the target of the proposed approach is coordinated model-free optimization of central chiller plants with multi-agents, the interference and competition between multiple agents needs to be considered. Hence, the value function of each agent is defined as $Q_i(s, a_i, a_{-i})$, where footnote i suggests the $i$th agent, $Q_i$ is the value function of the $i$th agent, $s$ is the system state defined before, $a_i$ is the action of the $i$th agent, $a_{-i}$ is the action of the other agent.

Specifically, the whole value function (including two agents' own value functions) in this study is designed as Table 1. The first row of Table 1 is its state space; each column represents a certain system state and its corresponding payoff matrix. The payoff matrix is comprised of chiller agent's actions (which is also called strategies in game theory), cooling agent's actions and their payoffs under each action pair. For example, (6.5, 0.8) means when the cooling agent's action is (30 Hz for cooling towers, 35 Hz for condenser water pumps) and the chiller agent's action is 9 °C, their expected payoffs (i.e., the value function values) are 6.5 and 0.8, respectively. This payoff matrix-based value function form is proposed for the convenience of the next step—equilibrium solving. Section 2.3 demonstrates how to find out the optimal control actions (cooling agent action and chiller

agent action) with the latest Table 1, and Section 2.4 demonstrates how to update this table based on environmental feedback (i.e., system reward).

### 2.3. Equilibrium Solving

As introduced in Section 2.2, in this study, the cooling agent takes system COP as the optimization objective, while the chiller agent optimizes both user comfort and system *COP* simultaneously. Due to the discrepancy between the objectives of these two agents, this study uses game theory approach to realize coordinated optimization. Specifically, the coordinated optimization procedure is regarded as a static bilateral game of complete information between two agents herein [34,35]. Each agent manages its own value function $Q_i(s, a_i, a_{-i})$ while both agents could observe the whole value function (the whole Table 1); in other word, two agents update Table 1 together. This method lets two agents observe each other's value function because they are more cooperative than competitive to each other. In other word, although their objectives are different, they do need to work together for the energy efficiency of the whole system. Hence, it would be not necessary for them to hide information from each other. Moreover, this complete information scheme could ease the following equilibrium solving process. To solve the equilibrium for two agents, the dominant strategy underlining method is adopted herein [34,35]:

(1) In the beginning of every optimization time step, the system state is observed, and a certain matrix related to the current state (e.g., Table 2) can be extracted from the whole value function table (i.e., Table 1). In Table 2, (6.5, 0.8) means at current state, if cooling agent takes (30 Hz, 35 Hz) as the next action and chiller agent takes 9 °C as the next action, then their expected payoff would be 6.5 and 0.8, respectively.

(2) Each agent underlines its optimal payoff for every potential action of the other agent. For instance (let us ignore the " ... " part for now for simplicity), cooling agent needs to underline 6.5 in Table 2 because if chiller agent takes 9 °C at current time step, the maximal payoff (i.e., value function value) for the cooling agent would be 6.5, in accordance with its optimal action (30 Hz, 35 Hz). Similarly, cooling agent needs to underline 6.8 and 6.6 in case that chiller agent takes 10 or 11 °C. On the other hand, for the chiller agent, it needs to underline 0.8 and 0.9 in Table 2 because, according to the current value function table (i.e., Table 2), no matter which action is taken by cooling agent, chiller agent should take 9 °C to maximize its own payoff.

**Table 2.** Example of the bilateral payoff matrix under a certain state *s*.

| $a_{cooling} \backslash a_{chiller}$ | **9 °C** | **10 °C** | **11 °C** |
|---|---|---|---|
| 30 Hz, 35 Hz | <u>6.5</u>, <u>0.8</u> | <u>6.8</u>, 0.7 | 6.3, 0.6 |
| ... ... | | ... ... | |
| 50 Hz, 50 Hz | 6.2, <u>0.9</u> | 6.4, 0.6 | <u>6.6</u>, 0.4 |

(1) After underlining all optimal payoffs, find cells with two lines. In Table 2, there is one cell corresponding to a strategy set of (30 Hz, 35 Hz, 9 °C), and this strategy set is a Nash equilibrium solution. If there are more than one strategy set, take the next step; otherwise the optimization of $\left(f_{tower}, f_{pump}, T_{chws}\right)$ is competed with the only answer.

(2) For matrix games with large strategy space (i.e., large action space of RL agents), there may be more than one Nash equilibrium solution. Under this circumstance, the proposed approach uses Pareto domination principle to refine the solutions [44]. Concretely, the proposed approach would compare all equilibrium solutions' payoffs; if one solution's payoff is dominated by anyone else, then this solution would be excluded. For instance, there are four solutions (i.e., four sets such as $\left(f_{tower}, f_{pump}, T_{chws}\right)$) corresponding to four payoffs (6.6, 0.8), (6.6, 0.9), (6.8, 0.8) and (6.7, 0.9). In this case, (6.6, 0.8) is dominated by the other three; (6.6, 0.9) is dominated by (6.7, 0.9); while (6.8, 0.8) and (6.7, 0.9) do not dominate each other. Hence the two strategy sets of payoffs, (6.6, 0.8) and (6.6, 0.9), would be **excluded** from the alternatives. After the

comparison above, **the other two** solutions remain as alternatives, and the solution with the maximal cooling agent payoff (which is (6.8, 0.8)) would be chosen as the optimal control action set; then the optimization of $\left(f_{tower}, f_{pump}, T_{chws}\right)$ is complete.

### 2.4. Value Function Update with SARSA

In every optimization time step, SARSA algorithm proposed by Rummery and Niranjan [45] is adopted herein to update agents' value functions (i.e., to update the values such as (6.5, 0.8) in Table 1 after solving the equilibrium of the matrix game:

$$Q_i(s, a_i, a_{-i}) \leftarrow Q_i(s, a_i, a_{-i}) + \alpha\left[r_i + \gamma Q_i\left(s', a'_i, a'_{-i}\right) - Q_i(s, a_i, a_{-i})\right] \tag{4}$$

where $Q_i(s, a_i, a_{-i})$ is the $i$th agent's value function value corresponding to the last state–action pair $(s, a_i, a_{-i})$, $s$ is the system state at the last time step, $a_i$ is the $i$th agent's action taken at the last time step, $a_{-i}$ refers to the last action taken by the other agent, $r_i$ is the latest reward received by the $i$th agent, $\alpha$ is the learning rate, $\alpha = 0.7$ in this study [13], $\gamma$ is the weight of expected future reward, $\gamma = 0.01$ according to Ref. [13]. $Q_i\left(s', a'_i, a'_{-i}\right)$ is the $i$th agent's value function value corresponding to current state $s'$ and the action plan $(a'_i, a'_{-i})$ solved in Section 2.3.

### 2.5. Hyperparameter Setting

This section addresses the setting of all static hyperparameters used in the proposed approach (i.e., excluding the variables monitored in real-time). In Equations (2) and (3), $T_{chwr, ref}$ is the referenced $T_{chwr}$ value (°C), and it is available on chiller manual; $\beta_1 = 0.25$, $\beta_2 = 4.15$, and they are coefficients determining the shape of comfort utility function curve (Figure 2); $COP_{s,nominal}$ is the quotient of rated system cooling capacity (kW) and rated system electrical power (kW); $X_{cop}$ is the weight of efficiency objective in chiller utility, which should be specified within 0–1 according to the need of the applied system, and it is set to 0.5 in this study.

In Equation (4), $\alpha$ is the learning rate, and $\alpha = 0.7$ in this study [13]; $\gamma$ is the weight of expected future reward, and $\gamma = 0.01$ herein according to Ref. [13].

## 3. Simulation Case Study
### 3.1. Virtual Environment Establishment

The control performance of the proposed method is evaluated via simulation. A real HVAC system along with its field data is adopted to establish a virtual central chiller plant for simulation. The layout of the case system is illustrated in Figure 3. System characteristics are listed in Table 3.

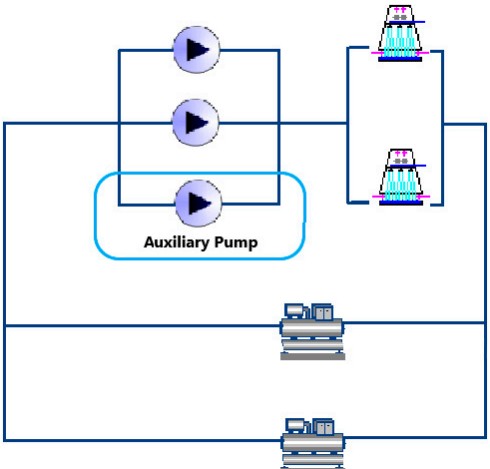

**Figure 3.** System layout. (The auxiliary pump is not included in the simulation herein.)

**Table 3.** Case system characteristics.

| Equipment | Number | Characteristics (Single Appliance) |
|---|---|---|
| Screw chiller | 2 | Cooling capacity = 1060 kW, power = 159.7 kW<br>Chilled water temperature = 10/17 °C<br>Chilled water flow rate = 131 m³/h (36.39 kg/s) |
| Condenser water pump | 2 + 1<br>(one auxiliary) | Power = 14.7 kW, flowrate = 240 m³/h<br>Head: 20 m, variable speed |
| Cooling tower | 2 | Power = 7.5 kW, flowrate = 260 m³/h, variable speed |

The central chiller plant model (Equations (5) and (6)) is established based on random forest [46] and field data from one cooling season of the real case system (from 1st June to 18th September 2019, with data sampling intervals of 10 min). Equation (5) models the real-time system electrical power $P_s$ (kW). Equation (6) models the temperature of the chilled water returning from the user side. Along with Equations (1)–(3), this central chiller plant model (i.e., Equations (5) and (6)) could provide environmental reward to RL agents for learning. Involved variables are introduced in Table 4. It is worth noting that the simulation based on the established system model is time-independent—the simulated system operation at each simulation time step does not influence adjacent time steps. In every time step, the system model receives the real-time environmental inputs ($CL_s$, $T_{wet}$) and control signals ($f_{pump}$, $n_{pump}$, $f_{tower}$, $n_{tower}$, $status_{chiller}$, $T_{chws}$); then it models the $P_s$ and $T_{chwr}$, feeding them back to RL agents; then its work is complete.

$$P_s = Random\ forest\left(CL_s, T_{wet}, f_{pump}, n_{pump}, f_{tower}, n_{tower}, status_{chiller}, T_{chws}\right) \quad (5)$$

$$T_{chwr} = T_{chws} + CL_s / \left(n_{pump} \times F_{chw} \times C_p\right) \quad (6)$$

**Table 4.** Nomenclature of involved variables.

| Variable | Description | Unit |
|---|---|---|
| $P_s$ | Real-time overall electrical power of chillers, condenser water pumps and cooling towers | kW |
| $CL_s$ | System cooling load | kW |
| $T_{wet}$ | Ambient wet-bulb temperature | °C |
| $f_{pump}$ | Common frequency of running condenser water pump(s) | Hz |
| $n_{pump}$ | Current working number of condenser water pumps (equal to the running number of chillers and chilled water pumps) | |
| $f_{tower}$ | Working frequency of running cooling tower(s) | Hz |
| $n_{tower}$ | Current working number of cooling towers | |
| $T_{chws}$ | Temperature of supplied chilled water | °C |
| $T_{chwr}$ | Temperature of returned chilled water | °C |
| $F_{chw}$ | Nominal chilled water flowrate of single chiller | kg/s |
| $C_p$ | Specific heat capacity of water | kJ/(kg·K) |
| $status_{chiller}$ | Current working status of chillers:<br>1–only Chiller 1 is running, 2–only Chiller 2 is running,<br>3–both chillers are running, 0–no chiller is running | |

Equation (6) is a simple physical process estimation equation while Equation (5) is a black-box model based on regression. Hence, the accuracy of the trained Equation (5) needs to be verified before we use it to model the operation of the case central chiller plant. The accuracy of the trained Equation (5) has been evaluated and verified in Ref. [5] with indicators including MAPE (mean absolute percentage error), CV-RMSE (coefficient of variation of the root mean square error) and $R^2$ (coefficient of determination). Details about modelling could be found in Ref. [5].

*3.2. Compared Control Algorithms*

In this study, the following control logics are adopted as comparative methods to evaluate the performance of the proposed approach.

***Basic control:*** This control logic keeps $f_{tower}$, $f_{pump}$, $T_{chws}$ at 50 Hz, 50 Hz and 10 °C, respectively (i.e., nominal values of these appliances).

***WoLF-PHC (Win or Learn Fast-Policy Hill Climbing) control:*** It is a classic MARL algorithm for the RL in non-fully cooperative multi-agent systems [47,48]. It was selected as the comparative algorithm for the following reasons:

(1) Typically the learning task in a MARL field can be categorized into fully cooperative task [49], fully competitive task [50] and non-fully cooperative (mixed) task. Among them, the mixed task is the most general and complicated task form. Different MARL algorithms could handle different types of tasks, WoLF-PHC is a universal algorithm that is capable of dealing with a non-fully cooperative (mixed) task [51], which is also the targeted problem in this study.

(2) WoLF-PHC can work feasibly in heterogeneous multi-agent systems. That is, even if not all agents are embedded with the same learning algorithm, WoLF-PHC could still work normally [51].

(3) WoLF-PHC does not require agent's prior knowledge about the task, which is the same as the approach proposed in this study [52].

(4) WoLF-PHC has been investigated for the multi-agent optimal control research in buildings and power grids [5,48,53], which proves its feasibility and performance.

In brief, WoLF-PHC is a feasible, classic and well-performing algorithm adaptable to the case problem in this study [51]. Hence, it is a suitable comparative algorithm to better evaluate and reveal the performance of the approach proposed herein. In this study, when applying WoLF-PHC, three agents are established: tower agent, pump agent and chiller agent. For tower agent, its action space is (30, 35, 40, 45, 50 Hz) for $f_{tower}$; the pump agent's action space is (35, 40, 45, 50 Hz) for $f_{pump}$; the chiller agent's action space is (9, 10, 11 °C) for $T_{chws}$. All agents share a common state space, which is the same as the proposed approach in Section 2.2. Moreover, the pump agent and tower agent share a common reward variable, system *COP*, as in Section 2.2; the chiller agent's reward variable is chiller utility, as in Section 2.2.

When applying WoLF-PHC, three functions need to be established for each agent: value function $Q_i(s, a_i)$, target policy $\pi_i(s, a_i)$ and historical average policy $\overline{\pi}_i(s, a_i)$ (footnote i refers to the *i*th agent). For each agent, all $Q_i(s, a_i)$ values are initialized to 0, all $\pi_i(s, a_i)$ values are initialized to $\frac{1}{|A_i|}$, all $\overline{\pi}_i(s, a_i)$ values are initialized to $\frac{1}{|A_i|}$ where $|A_i|$ is the size of the *i*th agent's action space (i.e., for pump agent, $|A_i| = 4$; for chiller agent, $|A_i| = 3$). The number of each state's occurrence is recorded by $C(s)$, and it is initialized to 0.

In the online control process, each agent updates its own $Q_i(s, a_i)$, $\overline{\pi}_i(s, a_i)$, $\pi_i(s, a_i)$ with Equations (7)–(13), where $Q_i(s, a_i)$ is the Q-value of the *i*th agent corresponding to the last state $s$ and its last action $a_i$; $\alpha$ is agents' learning rate, with $\alpha = 0.7$ referring to Ref. [13]; $r_i$ is the reward value of the *i*th agent from the last time step; $\gamma$ is the weight of the expected future reward, $\gamma = 0.01$ [13]; $\max_{a_i'} Q_i(s', a_i')$ is the maximum Q-value of the *i*th agent at cur-rent state $s'$; $\pi_i(s, a_{i,j})$ is the *i*th agent's target policy function value corresponding to the last state $s$ and the *j*th action $a_{i,j}$. Moreover, $\overline{\pi}_i(s, a_{i,j})$ is the *i*th agent's average policy function value corresponding to the last state $s$, and the *j*th action $a_{i,j}$; $\arg\max_{a_i'} Q_i(s, a_i')$ is the optimal action of the *i*th agent at the last state $s$. $\delta_{win}, \delta_{lose}$ are special parameters of WoLF-PHC, influencing how fast an agent adjusts its target policy. Referring to Refs. [52,53], $\delta_{lose} : \delta_{win} = 4$ is recommended considering convergence speed. In this study, different pairs of $\delta_{win}, \delta_{lose}$

are used to investigate the influence and potential risk of these two critical parameters: $\{(\delta_{win} = 0.01, \delta_{lose} = 0.05), (\delta_{win} = 0.03, \delta_{lose} = 0.15), (\delta_{win} = 0.05, \delta_{lose} = 0.25)\}$.

$$Q_i(s, a_i) \leftarrow Q_i(s, a_i) + \alpha \left[ r_i + \gamma \max_{a_i'} Q_i(s', a_i') - Q_i(s, a_i) \right] \tag{7}$$

$$C(s) = C(s) + 1 \tag{8}$$

$$\overline{\pi_i}(s, a_{i,j}) \leftarrow \overline{\pi_i}(s, a_{i,j}) + \frac{1}{C(s)} \left[ \pi_i(s, a_{i,j}) - \overline{\pi_i}(s, a_{i,j}) \right] \; for \; \forall a_{i,j} \in A_i \tag{9}$$

$$\pi_i(s, a_{i,j}) \leftarrow \pi_i(s, a_{i,j}) + \Delta_{s, a_{i,j}} \qquad for \; \forall a_{i,j} \in A_i \tag{10}$$

where:

$$\Delta_{s, a_{i,j}} = \begin{cases} -\delta_{s, a_{i,j}} & if \; a_{i,j} \neq \underset{a_i'}{argmax} Q_i(s, a_i') \\ \sum_{a_{i,c} \neq a_{i,j}} \delta_{s, a_{i,c}} & else \end{cases} \tag{11}$$

where:

$$\delta_{s, a_{i,j}} = \min \left( \pi_i(s, a_{i,j}), \frac{\delta}{|A_i| - 1} \right) \tag{12}$$

$$\delta = \begin{cases} \delta_{win} & if \; \sum_{a_{i,j} \in A_i} \pi_i(s, a_{i,j}) Q_i(s, a_{i,j}) > \sum_{a_{i,j} \in A_i} \overline{\pi_i}(s, a_{i,j}) Q_i(s, a_{i,j}) \\ \delta_{lose} & else \end{cases} \tag{13}$$

## 4. Results and Discussion

To verify the performance of the proposed control approach, simulations are conducted via the virtual environment established before. The operation of the case system over a cooling season (from 1 June to 18 September 2019) is simulated under five different controllers (Table 5). In each step of simulations, one data point of the measured $T_{wet}$, $CL_s$ are utilized as environmental inputs to the controller and the system model; after receiving the inputs, the controller would determine the forthcoming control plan to the system model; based on environmental inputs and control signals, the system model could output the modelled system power and $T_{chwr}$, which would be referred by the controller for reinforcement learning [5].

**Table 5.** Simulation cases.

| Case | Controller Algorithm | Cooling Tower Action | Condenser Pump Action | Chiller Action | Parameters | State | Reward |
|---|---|---|---|---|---|---|---|
| 1 | Baseline | 50 Hz | 50 Hz | 10 °C | / | / | / |
| 2 | WoLF-PHC | 30, 35, 40, 45, 50 Hz | 35, 40, 45, 50 Hz | 9, 10, 11 °C | $\alpha = 0.7$ $\gamma = 0.01$ $\boldsymbol{\delta_{win} = 0.01}$ $\delta_{lose} = 0.05$ | $T_{wet}$, $CL_s$ | System $COP$, chiller utility |
| 3 | | | | | $\alpha = 0.7$ $\gamma = 0.01$ $\boldsymbol{\delta_{win} = 0.03}$ $\delta_{lose} = 0.15$ | | |
| 4 | | | | | $\alpha = 0.7$ $\gamma = 0.01$ $\boldsymbol{\delta_{win} = 0.05}$ $\delta_{lose} = 0.25$ | | |
| 5 | Game theory MARL | Jointed action-like (pump 50 Hz, tower 30 Hz) | | 9, 10, 11 °C | $\alpha = 0.7$ $\gamma = 0.01$ | | |

As listed in Table 5, five controllers are generated for a comparative simulation case study: one baseline controller which keeps $f_{tower}, f_{pump}, T_{chws}$ at nominal values; three WoLF-PHC controllers with different pairs of ($\delta_{win}$, $\delta_{lose}$); one Game MARL controller in accordance with our proposed approach. Moreover, due to the stochastic nature of RL algorithms, Case 2–5 are all repeated for three times independently, and the averaged results of each case are analyzed herein. In addition, Case 2–5 do not include pre-training, which means all agents are initialized right before simulation. Table 5 suggests that the proposed Game MARL controller has fewer pending parameters than WoLF-PHC does, denoting its robustness in engineering applications.

### 4.1. First Cooling Season Performance

Simulation results over the first cooling season are addressed in Figure 4 and Table 6. Figure 4 illustrates distributions of system *COP*, chiller utility and $T_{chwr}$ in the first simulated cooling season. In Figure 4, WoLF_0.01 corresponds to Case 2, WoLF_0.03 corresponds to Case 3, WoLF_0.05 corresponds to Case 4.

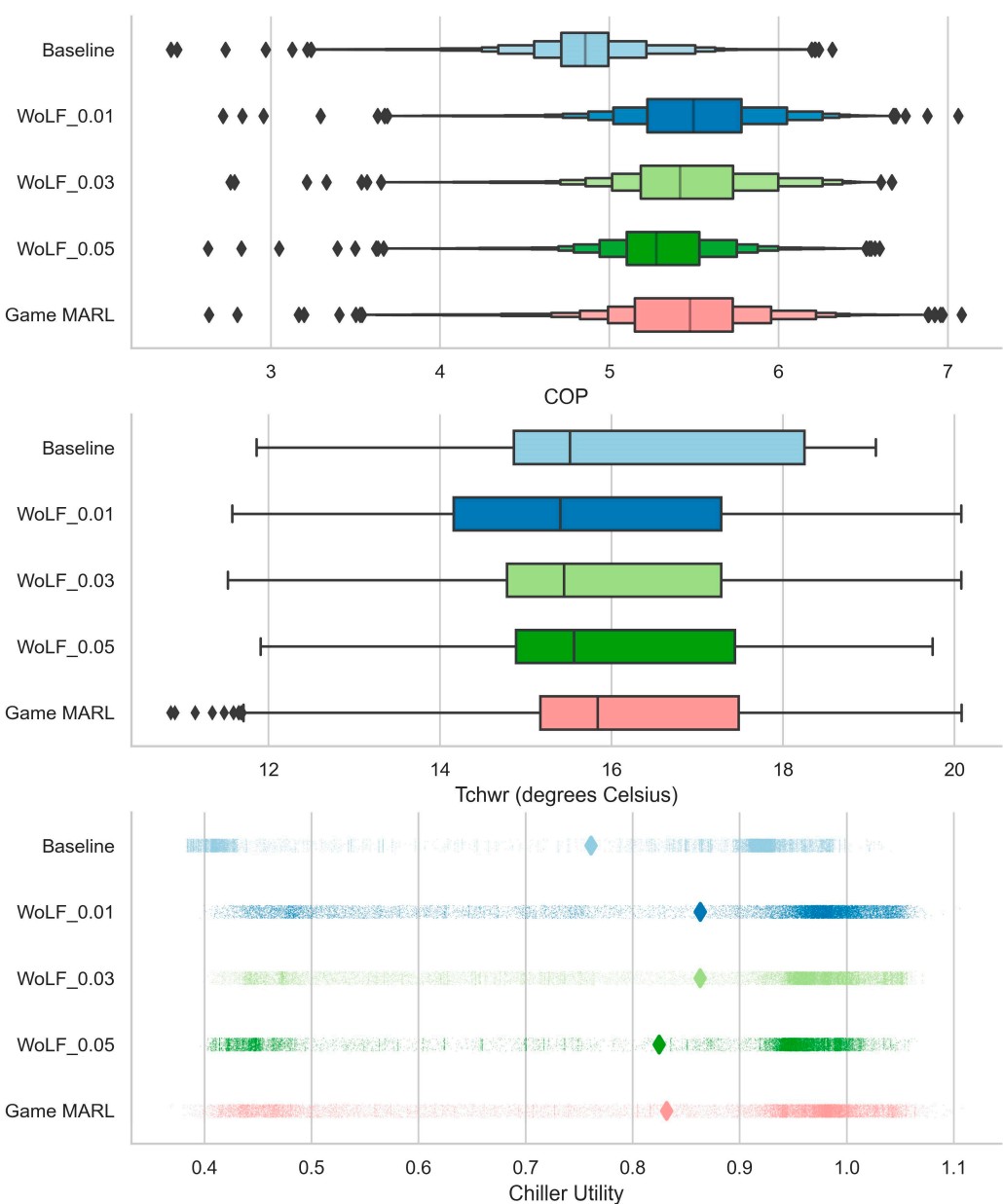

**Figure 4.** Distributions of the system *COP*, $T_{chwr}$ and chiller utility under each controller.

**Table 6.** Simulation results on the first episode (first cooling season).

| Case | Total energy Consumption (kWh) | Cumulated Chiller Utility |
|:---:|:---:|:---:|
| 1 | 549,101 | 11,511.58 |
| 2 | 486,865 | 13,055.17 |
| 3 | 490,685 | 13,052.78 |
| 4 | 505,180 | 12,474.10 |
| 5 | 491,623 | 12,579.88 |

Distributions of the system *COP* indicate that the central boxes of Game MARL controller and WoLF_0.01 controller are higher than WoLF_0.05 and the baseline. Moreover, median values (i.e., the line at the middle of the central box) show the same. Therefore, generally, the Game MARL controller and WoLF_0.01 controller perform well on enhancing system energy efficiency. To this fact, deduced reasons include the following: (1) the basic controller keeps all appliances running at nominal operational values, wasting energy conservation potential; (2) the two critical parameters of WoLF-PHC (i.e., $\delta_{win}, \delta_{lose}$) evidently influence the learning effectiveness of the WoLF-PHC algorithm, and misspecification of these two parameters (e.g., WoLF_0.05) weakened the performance of this algorithm in this case study [52].

About the results of $T_{chwr}$, it is observed that the general $T_{chwr}$ distribution of the Game MARL controller is relatively higher than the others', denoting its aggressive strategy on adjusting the chillers' $T_{chws}$ set point (because tunning up $T_{chws}$ could enhance chiller efficiency but may sacrifice user comfort [54]). This fact (higher $T_{chwr}$ of Game MARL) is also corresponding to the chiller utility distribution, where Game MARL's conditional mean value (i.e., the rhombus point on the scatter swarm) is lower than those of WoLF_0.01 and WoLF_0.03 because Game MARL lost some comfort utility. Moreover, regarding the chiller utility distribution, the conditional mean value of WoLF_0.05 is lower than Game MARL's although its $T_{chwr}$ is lower (which is better for the comfort utility objective). That is because WoLF_0.05's advantage on user comfort does not neutralize its shortage on system efficiency.

Table 6 lists the quantified results of each case. It suggests that WoLF_0.01 performs the best on saving energy and maintaining user comfort; Game MARL's performance is close to WoLF_0.03, better than WoLF_0.05. Table 6 also verified the deduction before; misspecification of $\delta_{win}, \delta_{lose}$ weakens the performance of WoLF-PHC, which brings potential risk to this algorithm.

It is also worth noting that the Case 1 energy consumption is not equal to the result in Ref. [5] because the baseline controller herein adopts a constant 10 °C $T_{chws}$ set point, while it is 11 °C in Ref. [5]. Furthermore, $T_{chws}$ influences the chillers' energy consumption.

*4.2. Performance Evolution in Five Cooling Seasons*

For RL algorithms, the post-convergence performance is as important as the initial short-term performance. In addition, the RL agent's convergence speed is another critical indicator for engineering application.

Figure 5 shows the long-term performance of the Game MARL controller and the WoLF_0.01 controller in continuous five-episode simulations (no pre-training). Solid lines are the energy saving rate of two controllers compared to the baseline controller. Solid and dashed lines suggest that both RL-based controllers' performance converge at the second episode (i.e., the second cooling season); this time cost is acceptable for engineering practices. WoLF_0.01 outperforms Game MARL in cumulated chiller utility value by about 4%, which proves that WoLF-PHC, with an appropriate hyper-parameter setting, is a mature and feasible MARL algorithm for central chiller plants' optimal control. Meanwhile, their post-convergence energy saving rates are very close (gap less than 0.5%), which validates the energy saving performance of the proposed Game MARL approach. Moreover, our approach's nature of fewer pending parameters implies its robust anticipated effectiveness and feasibility for engineering application.

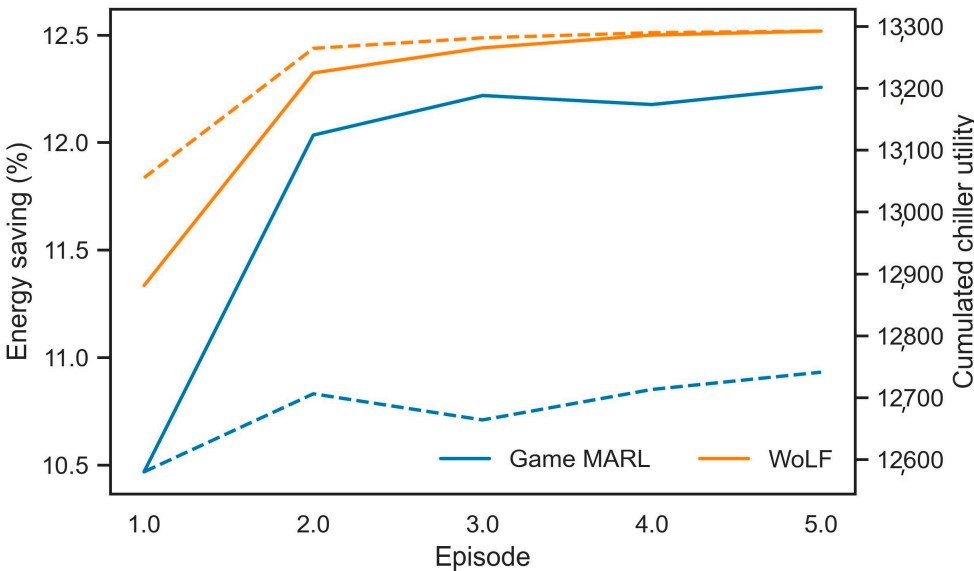

**Figure 5.** Long-term evolution of two MARL controllers.

## 5. Conclusions and Future Work

### *5.1. Conclusions*

Optimal control of central chiller plants has been widely investigated due to its great importance for building energy conservation and decarbonization. Model-free control based on RL techniques is promising for engineering practices thanks to its avoidance of model uncertainty risk [17]. However, the popular single RL agent scheme suffers from a slow converging speed and a long training period when optimizing multiple controlled variables for central chiller plants [5,26]. Multi-agent scheme is suitable for the simultaneous optimization of multiple controllable variables, but it also faces the challenge of how to coordinate multiple agents' local optimization procedure. Moreover, previous decentralized control frameworks for a multi-agent scheme are mostly designed for model-based control algorithms. Therefore, it is necessary to develop a multi-agent model-free optimal control approach for the optimization of central chiller plants.

To realize that goal, this paper proposes a multi-agent model-free optimization approach for central chiller plants using multi-agent reinforcement learning and game theory. The proposed approach describes the coordinated problem as a bilateral matrix game between cooling side appliances and chillers considering their inconsistent operational objectives. Meanwhile, this approach adopts SARSA as the RL algorithm for agents' learning. Different to previous research, the method proposed herein has the following features simultaneously: model-free optimization; multi-agent optimization scheme; consideration of the objective inconsistency of involved agents. The abovementioned characteristics imply the considerable feasibility of the proposed method in engineering practices.

A data-driven system model is established with measured field data of a real HVAC system. The system model is taken as the virtual environment for simulation case studies. Five different controllers are generated to control the virtual system. Results indicate that, at the initial stage (the first cooling season without pre-training), the proposed approach could save over 10% of system energy compared to baseline, which is acceptable and close to the classic MARL algorithm, WoLF-PHC. After the first cooling season, the performance of the proposed controller nearly converged at a 12% energy savings rate. Simulation results proved that compared to WoLF-PHC, the proposed approach's nature of fewer pending parameters implies its robust anticipated effectiveness, validating its feasibility for engineering practices.

*5.2. Future Work*

As addressed in Section 2.1, the proposed approach is designated to optimize the operation of chillers, cooling towers and condenser water pumps. Chilled water pumps are not included in the optimization herein. Hence, in the future, the coordinated model-free optimization including all appliances above is worth investigation to better utilize the energy saving potential of central chiller plants.

In addition, although the proposed approach is embedded with fewer parameters than WoLF-PHC (which leads to less parameter tuning work), the calculation process of the proposed method is more complicated for comprehension. It would be meaningful to simplify its workflow for engineering practice in the future.

The proposed method adopts a tabular RL algorithm which requires discretization of state and action variables, and the influence of the discretization granularity to control performance is worth quantitative analysis in the future.

**Author Contributions:** Conceptualization, S.Q.; Methodology, S.Q. and Z.P.; Software, S.Q.; Validation, S.Q. and Z.P.; Formal analysis, Y.T.; Investigation, Z.P.; Resources, Z.L. (Zhengwei Li); Data curation, Y.T.; Writing—original draft, S.Q.; Writing—review and editing, S.Q.; Visualization, S.Q.; Supervision, Y.T., Z.L. (Zhengwei Li) and Z.L. (Zhenhai Li); Project administration, Z.L. (Zhenhai Li) and Z.L. (Zhengwei Li). All authors have read and agreed to the published version of the manuscript.

**Funding:** This research received no external funding.

**Institutional Review Board Statement:** Not applicable.

**Informed Consent Statement:** Not applicable.

**Data Availability Statement:** Data sharing not applicable.

**Conflicts of Interest:** The authors declare no conflict of interest.

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
