# Peer review of "Multi-Agent Optimal Control for Central Chiller Plants Using Reinforcement Learning and Game Theory"

_systems, doi:10.3390/systems11030136_

Round 1

Reviewer 1 Report

The manuscript is innovative in using reinforcement learning and game theory for multi-agent optimal control of central chiller plants. However, the approach studied in the manuscript is too complex and the present study may be an interesting one for scholars in the field; it is difficult for managers to understand in practical industrial applications. The manuscript has some minor errors in English; there are some parts of the structure that need improvement. Specific recommendations are as follows:

1. The entire introduction looks more like a literature review, and the authors are advised to revise the structure of the entire introduction. For example, a brief background of this study could be given before presenting the present content.

2. The manuscript suddenly appears "Step3" and "Step2" in lines 87 and 88 of the introduction, which may confuse the reader. Suggest adding "step1,step2,step3" when describing "three steps".

3. Regarding the model-free optimization problem for the whole central chiller plant, there are two problems remaining: (1) the single RL agent scheme is incapable of optimizing multiple variables in the whole central chiller plant due to the enormous action space; and (2) there are potential objectives conflicting between multiple appliances that need reconciliation.” Would it be better to change the sentence in lines 138-142 of the manuscript introduction to this?

4. it is suggested that the authors should give a more detailed discussion about multi-agent optimization   , such as previous relevant studies, t: https://doi.org/10.56578/jimd010108; https://doi.org/10.56578/jimd010103.

4. Please double-check the sentences in lines 258-261 for grammatical errors.

5. The authors are invited to check carefully whether the content of lines 291-298 in the manuscript can be interpreted as point (5).

6. In line 328 of the manuscript, "The model" refers to which of the "models" described in the preceding text, and the author should explain.

7. The formulas in lines 367-373 of the manuscript are messy, and the authors are advised to re-edit and reformat them for readability.

8. The font of the figure title in 400 lines of the manuscript is obviously different from the other figures, so please check carefully and revise it carefully.

9. There is no "future work" in the "conclusion and future work" section. Please add "future work".

10. Authors are invited to double-check that the submitted manuscript is complete and why author contributions, etc., are not provided.

In summary, it is recommended that the authors make major revisions to the manuscript.

Author Response

Thank you very much for your review. Please check the attached response letter.

Reviewer 2 Report

The article addresses the problem of the optimization of central chiller plants using a multi-agent approach using game theory and reinforcement learning, the results are compared to the classic WoLF-PHC algorithm. here some comments

·        In the introduction, it is recommended to broaden the discussion of the model free techniques adopted by reinforcement learning in the study of chillers, since only references are mentioned [11,13-15].

·        a minor grammar check is recommended to correct various misspellings

·        Although the contribution of the article to knowledge is evident, I recommend that the main contributions of the proposed method and its differences with the proposals found in the literature be declared promptly.

·        I recommend that the proposed pseudocode be shown in the methodology, as well as the parameters used.

·        improve the format of equations 7-13

·        In the results, the authors should explain why the WoLF-PHC algorithm was chosen to compare the proposed method.

Author Response

Thank you very much, please check the attached response letter.

Reviewer 3 Report

This article focuses on efficiently and optimally utilizing central chiller plants while tackling the issues of model uncertainty and combining the advantages of MARL and game theory to achieve an optimal dominant solution for the chiller plant controller coordination and learning.

The article is well written, easy to follow, however the assumptions for the formulation are too simplistic and needs significant modification when applied to real world applications. Nevertheless, the methodology and experimentation design layout a step in the right direction for this very important and challenging problem in the energy efficiency space. Please find my comments below.

1. One of the motivations cited by the authors for using MARL framework as compared to using RL frameworks is that "single RL agent scheme is incapable of optimizing multiple variables in the whole central chiller plant due to the enormous action space".

However, I disagree with how this motivation has been used. Authors have not justified why the particular case they are solving suffers from curse of dimensionality. What are the challenges of discretizing the space too coarsely. For ex, authors themselves have used an integer variable for the Chiller Water temperature, thereby already discretizing the action and state space. Hence, the advantages of Multi-agent RL needs more justification and clarification for this article in terms of computation and space complexity comparisons.

To further add to this, the following study:

"Lazic, Nevena, Craig Boutilier, Tyler Lu, Eehern Wong, Binz Roy, M. K. Ryu, and Greg Imwalle. "Data center cooling using model-predictive control." Advances in Neural Information Processing Systems 31 (2018)"

successfully uses vanilla RL frameworks to control large number of chiller plants in a real-world Data Center cooling plant. 

2. Identical chillers and identical cooling towers assumptions are too simplistic. Chillers can have different thermal time constants that can cause the algorithm to be stuck in a "fighting mode" while trying to optimize for efficient set points and operating conditions for each chiller when they are not identical, thereby sacrificing on energy efficiency. How will this work handle such conditions? What would be the utility function for the different chillers and cooling towers in such a case?

3. What is the timescale of dispatching set points to the chiller and cooling tower, pump controllers? Seconds, minutes or hours? What is the impact of doing faster control vs slower controller on the optimization? A detailed analysis and experimental verification of this is warranted to understand the impact of proposed method on the dynamical system.

4. XCOP = Constant (0.5) in the study is not accurately justified as XCOP should be dependent on external factors like wet-bulb and dry-bulb temperatures, Relative humidity, altitude etc. that can cause effective chiller thermal capacity ratings to vary significantly, for ex. Psychometric curves. This should be incorporated in the design.

Moreover, it is not clear how certain unreachable states are dealt with in the SARSA and Nash Equilibrium formulation as certain higher ambient temperatures can cause chillers to not effectively reach a certain return temperature for a given ftwr, fpump or vice-versa.

5. In eq (5), why is the physics based model for chiller plants not adopted as compared to random forest based model. This model is well known in the mechanical system design.

6. Water loss in the system due to evaporative cooling and higher ftower should be considered a factor in the design. 

Author Response

Thank you for your suggestions and comments, please check the reponse letter.

Round 2

Reviewer 1 Report

The authors have revised the paper well according to my suggestion

Reviewer 3 Report

I thank the authors for careful consideration of this reviewer's comments. Authors have taken a thorough look into the comments and questions posed and have responded with comprehensive literature and insightful engineering clarifications.

This reviewer has no further comments.